# *Drosophila* R8 photoreceptor cell subtype specification requires *hibris*

Hong Tan[1¤a], Ruth E. Fulton[1], Wen-Hai Chou[2¤b], Denise A. Birkholz[1], Meridee P. Mannino[1], David M. Yamaguchi[1], John C. Aldrich[3], Thomas L. Jacobsen[3], Steven G. Britt[3]*

**1** Department of Cell and Developmental Biology, School of Medicine, University of Colorado Anschutz Medical Campus, Aurora, Colorado, United States of America, **2** Department of Molecular Medicine, University of Texas Health Science Center, San Antonio, Texas, United States of America, **3** Department of Neurology, Department of Ophthalmology, Dell Medical School, University of Texas at Austin, Austin, Texas, United States of America

¤a Current address: Department of Pathology, School of Basic Medical Sciences, Xi'an Jiaotong University Health Science Center, Xi'an, China
¤b Current address: Center for Neuropsychiatric Research, National Health Research Institutes, Miaoli, Taiwan
* steve.britt@austin.utexas.edu

**Data Availability Statement:** All relevant data are within the manuscript and its Supporting Information files.

## Abstract

Cell differentiation and cell fate determination in sensory systems are essential for stimulus discrimination and coding of environmental stimuli. Color vision is based on the differential color sensitivity of retinal photoreceptors, however the developmental programs that control photoreceptor cell differentiation and specify color sensitivity are poorly understood. In *Drosophila melanogaster*, there is evidence that the color sensitivity of different photoreceptors in the compound eye is regulated by inductive signals between cells, but the exact nature of these signals and how they are propagated remains unknown. We conducted a genetic screen to identify additional regulators of this process and identified a novel mutation in the *hibris* gene, which encodes an *irre* cell recognition module protein (IRM). These immunoglobulin super family cell adhesion molecules include human KIRREL and nephrin (NPHS1). *hibris* is expressed dynamically in the developing *Drosophila melanogaster* eye and loss-of-function mutations give rise to a diverse range of mutant phenotypes including disruption of the specification of R8 photoreceptor cell diversity. We demonstrate that *hibris* is required within the retina, and that *hibris* over-expression is sufficient to disrupt normal photoreceptor cell patterning. These findings suggest an additional layer of complexity in the signaling process that produces paired expression of opsin genes in adjacent R7 and R8 photoreceptor cells.

## Introduction

Color vision in humans and most other organisms is dependent upon the expression of spectrally distinct visual pigments (opsins) in different photoreceptor cells [1–3]. The organization of photoreceptor cells within the retinal mosaic reflects a variety of different developmental mechanisms, including regional specialization, stochastic, and precise cell-cell adjacency [4]. *D. melanogaster* is capable of color vision and is a useful experimental system for examining

**Funding:** S.G.B. received National Institutes of Health, National Eye Institute grant R01EY018376 in support of this work. https://www.nei.nih.gov The funder played no role in the study design, data collection and analysis, decision to publish, or preparation of the manuscript.

**Competing interests:** The authors have declared that no competing interests exist.

the developmental programs that produce photoreceptor cells having different color sensitivities [5–12]. The compound eye consists of ~800 ommatidia, each containing eight rhabdomeric photoreceptor cells (R cells). The central R7 and R8 photoreceptor cells mediate polarization sensitivity and color vision [13, 14]. As shown in **Fig 1**, the majority of ommatidia contain matched pairs of R7 and R8 cells expressing specific rhodopsin (Rh) visual pigments, either *Rhodopsin 3* (*Rh3*, FBgn0003249) and *Rhodopsin 5* (*Rh5*, FBgn0014019) (tandem magenta-blue cylinders), or *Rhodopsin 4* (*Rh4*, FBgn0003250) and *Rhodopsin 6* (*Rh6*, FBgn0019940) (tandem yellow-green cylinders).

These two main ommatidial subtypes were initially identified based on pale or yellow fluorescence when illuminated with blue light [15, 16], with pale (R7p/R8p) expressing *Rh3/Rh5*, while yellow (R7y/R8y) cell pairs express *Rh4/Rh6* (**Fig 1**) [10, 11, 17]. This paired expression of opsin genes in adjacent R7 and R8 cells within an individual ommatidium is thought to result from a series of developmental steps. First, a subset of R7 cells stochastically and cell autonomously express *spineless* (*ss*, FBgn0003513) which represses *Rh3* and induces *Rh4* expression [18–21]. In R7p cells that stochastically fail to express *ss* and do express *Rh3*, a signal is initiated that induces the expression of *Rh5* in adjacent R8p cells. Extensive studies have identified the genes *warts (wts*, FBgn0011739*)*, *melted (melt*, FBgn0023001*)*, members of the *hippo (hpo*, FBgn0261456*)* pathway, along with the TGFβ superfamily receptors *baboon* (*babo*, FBgn0011300) and *thick vein* (*tkv*, FBgn0003726), their respective ligands and numerous transcription factors as components of the induced versus default signal that establishes R7 and R8 photoreceptor cell subtype patterning [8, 12, 22–26]. This signal from R7p drives the expression of *Rh5* in R8p, and in the absence of a signal from R7y, the default R8y fate and expression of *Rh6* occurs. In addition, we have found that the *Epidermal growth factor receptor (Egfr,* FBgn0003731*)* and *rhomboid (rho,* FBgn0004635*)* are also required for this process [27, 28].

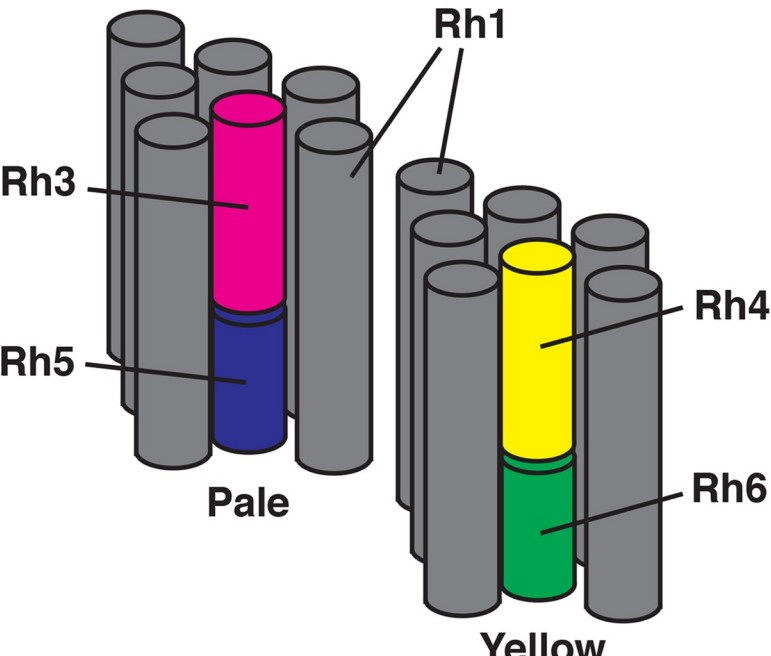

**Fig 1. Diagram of photoreceptor cell organization and opsin gene expression.** Two ommatidia are shown consisting of gray cylinders corresponding to the rhabdomeres of the R1-6 photoreceptor cells, which express Rh1. These surround the central rhabdomeres of the R7 and R8 cells. Expression of opsin genes within the R7 cells (*Rh3* in magenta or *Rh4* in yellow) is paired with opsin gene expression in the adjacent R8 cell (*Rh5* in blue or *Rh6* in green) in pale and yellow ommatidia, respectively.

Here we undertook a genetic screen to identify additional genes required for this process and show that *hibris* (*hbs*, FBgn0029082), an *irre* Cell Recognition Molecule (IRM) [29], NPHS1 (nephrin, *Homo sapiens*, HGNC:9801*)* related member of the Immunoglobulin Super Family (IgSF), is required for the establishment of paired opsin expression in adjacent R7 and R8 photoreceptor cells. We found that *hbs* is required within the retina for this process, suggesting that it interacts with the network of genes that regulate R7 and R8 photoreceptor cell differentiation.

## Results

### Isolation and characterization of the *a69* mutant

To identify genes required for the induction of *Rh5* expression in R8 photoreceptors, we screened approximately 150 homozygous viable eye-expressing enhancer trap lines carrying insertions of the *P{etau-lacZ}* transposon (FBtp0001352) [30]. This was based on the rationale that genes required for the induction of *Rh5* expression would be expressed in the eye, the *P{etau-lacZ}* transposon has been especially useful in studies of the nervous system, and insertion of this element into loci of interest would provide a convenient means to identify the affected genes [30]. The percentage of *Rh5*-expressing R8 cells was determined by labeling dissociated ommatidia with antibodies against *Rh5* and *Rh6*. Several mutants with abnormal percentages of *Rh5*-expressing R8 cells were noted and *a69* (FBgn0026612), with the lowest percentage of *Rh5* (9%) was further characterized. Immunostaining of both dissociated ommatidia and tissue sections showed that in the *a69* enhancer-trap line, *Rh5*-expressing R8 cells are reduced and most R8 cells have assumed the default fate and express *Rh6* (**Fig 2A, 2B, 2D and 2E**, **Table 1**). Since mutants lacking R7 cells or having a reduced number of *Rh3* expressing R7

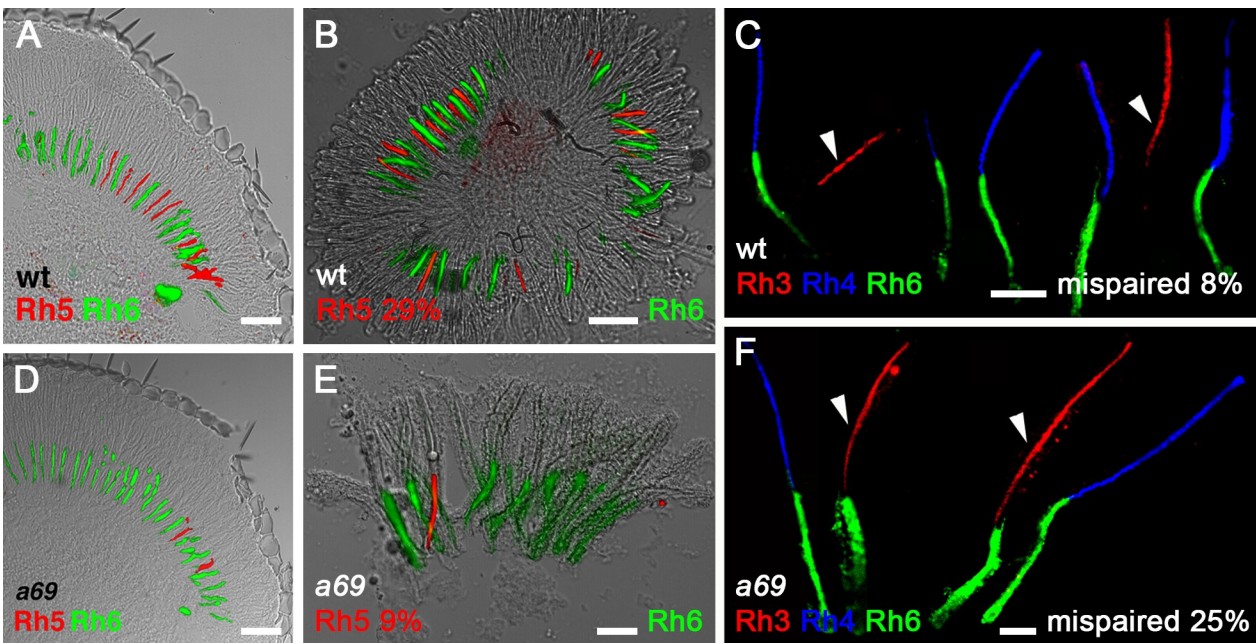

**Fig 2. *a69* mutants have a defect in *Rh5* and *Rh6* expression in R8 photoreceptor cells.** White eyed wild type (wt) flies (*w^1118^*) express *Rh5* and *Rh6* in a proportion of approximately 1:2 (*Rh5* 29%), this is shown in a longitudinal section of the retina (**A**) as well as in dissociated ommatidia (**B**). (**C**) wt flies express *Rh4* and *Rh6* in a paired fashion. The arrowheads indicate Rh3 expressing R7 cells, which are only rarely mispaired with *Rh6* expressing R8 cells (~6–8%). *w^1118^*; *P{etau-lacZ}a69* mutants show a disruption in *Rh5* expression, with a substantial decrease in *Rh5* expression (*Rh5* 9%) shown in both section (**D**) and dissociated ommatidia (**E**) as well as prominent mispairing between *Rh3* expressing R7 cells and *Rh6* expressing R8 cells in the same ommatidia (**F**) (arrowheads, mispaired 25%). Specific quantification and statistics are located in **Table 1**. Scale bars in each panel correspond to **A** 50µm, **B** 50µm, **C** 25µm, **D** 50µm, **E** 25µm, and **F** 10µm.

cells would also show diminished *Rh5* expression, we next examined the R7 cells and found that the percentage of those expressing *Rh3* was similar to *white*[1118] (*w*[1118], RRID:BDSC_3605) control flies (42% vs. 47%, **Table 1**). However, there was a dramatic increase in mispairing between *Rh3* expressing R7 cells adjacent to *Rh6* expressing R8 cells (**Fig 2C and 2F**, **Table 1**) compared to both *w*[1118] and *cinnabar*[1] *brown*[1] controls (*cn*[1] *bw*[1], RRID:BDSC_264) consistent with the idea that the *a69* enhancer trap line carries a mutation in a gene required for the induction of *Rh5* expression in R8 cells.

To isolate the gene responsible for the *a69* phenotype, the location of the P-element insertion in *a69* was determined and found to map to the right arm of the second chromosome at position 60E. To determine whether the P-element in *a69* is the cause of the phenotype, P-element excision lines were generated and analyzed. Thirty-five homozygous strains of these excision chromosomes were analyzed by staining dissociated ommatidia with antibodies against *Rh5* and *Rh6*, and all of them (100%) were found to have a low *Rh5* percentage, similar to that of *a69*. Only 1% of excision strains would be expected to retain the mutant phenotype as a result of imprecise excision, thus our inability to revert the mutant phenotype is consistent with the *a69* P-element not being responsible for the mutation [31]. Furthermore, mapping via recombination analysis revealed that the *a69* mutation is localized to the interval between the *purple* (*pr*, FBgn0003141) and *curved* (*c*, FBgn0000245) genes in the middle of the second chromosome (**Fig 3**, **S1 Table**), far away from the P-element insertion site in *a69*. From this we conclude that the *a69* mutation is not associated with the insertion of the P-element. Thirty-three deficiency lines located in the region between *pr* and *c* were tested for *a69* complementation (**S2 Table**). These analyses narrowed the location of the *a69* mutation to 51C3-51D1 on the right arm of the second chromosome (**Fig 4**). The lower portion of **Fig 4** shows a diagram of this genomic region, spanning ~300 Kb and encompassing 25 known protein coding genes.

**Table 1. Opsin expression in different genetic backgrounds.**

| Genotype | R8 cells expressing *Rh5*% (n) | R7 cells expressing *Rh3*% (n) | Mis-pairing | Figure |
|---|---|---|---|---|
| | | | % (n) | |
| *w*[1118] | 29 (214) | 47 (362) | *Rh3/Rh6* 8 (169) | 2A, 2B, 2C |
| | | | *Rh4/Rh5* 0 (424) | |
| *a69* | 9 (335) | 42 (241) | *Rh3/Rh6* 25 (253) | 2D, 2E, 2F |
| | SDF *w*[1118], $p = 1.9 \times 10^{-9}$ | | SDF *cn*[1] *bw*[1], $p = 1.2 \times 10^{-8}$ | |
| | | | SDF *w*[1118], $p = 1.2 \times 10^{-5}$ | |
| | | | *Rh4/Rh5* 0 (315) | |
| *cn*[1] *bw*[1] | ND | ND | *Rh3/Rh6* 6 (240) | |
| *GMR-hbs* | 70 (553) | 37 (445) | ND | 8A |
| | SDF *w*[1118], $p < 10^{-15}$ | SDF *w*[1118] | | |
| | | $p = 0.006$ | | |
| *sev*[14]; | 51 (1617) | R7 cells absent | NA | 8B |
| *GMR-hbs* | SDF *GMR-hbs*, $p < 10^{-15}$ | | | |
| | SDF *w*[1118]*sev*[14], $p < 10^{-15}$ | | | |
| *w*[1118] *sev*[14] | 12 (585) | R7 cells absent | NA | |
| | SDF *w*[1118], $p = 1.9 \times 10^{-9}$ | | | |

Statistical comparisons of strains were carried out as described in the Methods; n = the number of ommatidia counted. Unless indicated, the observed percentages were not significantly different from *w*[1118]. Strains compared to another control are indicated. Abbreviations are as follows: Significantly Different From (SDF) the strain indicated, at the *p* value shown by a two tailed test; Not Determined (ND); Not Applicable (NA).

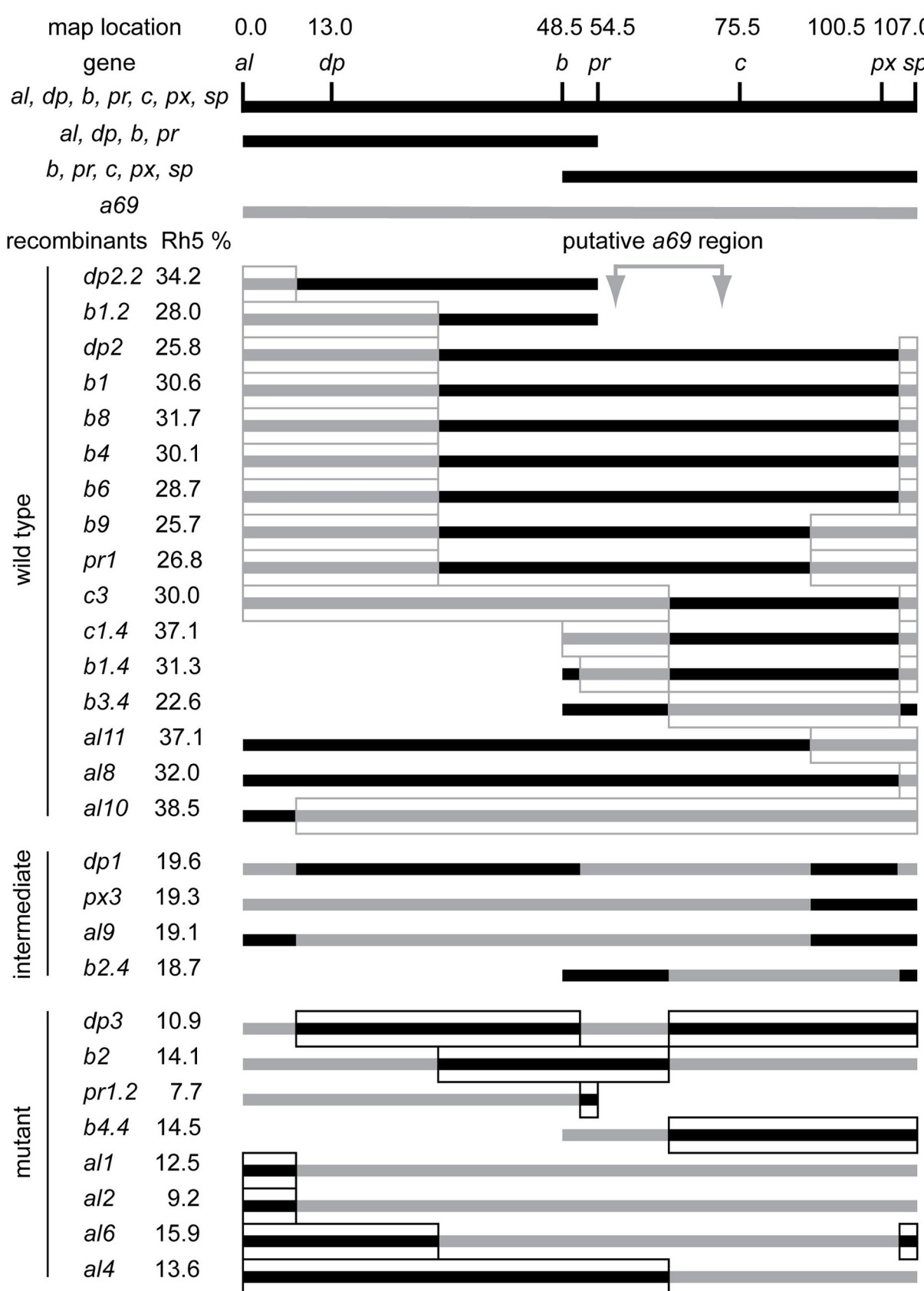

**Fig 3. Recombination mapping of *a69* to the second chromosome between *pr* and *c*.** Three multiply marked chromosomes (*al¹ dpy^{ov1} b¹ pr¹ c¹ px¹ sp¹*, *al¹ dpy^{ov1} b¹ pr¹*, and *b¹ pr¹ c¹ px¹ sp¹*) were recombined with the *w^{1118}*; *P{etau-lacZ}a69* mutant. After marker identification, recombinant strains were back crossed to the *a69* mutant and scored for the percentage of *Rh5* expression. The regions of the recombinant chromosomes assumed to be derived from the *a69* parental mutant strain are indicated in gray, while the regions assumed to be derived from the multiple marked (wild-type) chromosomes are black. Sixteen recombinant strains were phenotypically wild-type and complemented *a69*. Four recombinant strains were intermediate and eight strains were mutant and failed to complement *a69*. The four intermediate strains and one wild type strain, *al10*, differed from the expected phenotypes and may have resulted from multiple recombination events or exposure of cryptic modifier loci. See **S1 Table**. Complementation of *a69* Recombinant Strains.

To identify the gene specifically affected in the *a69* mutation, we took two approaches. First, a subset of genes were examined for alterations in expression in the *a69* mutant, and second, a large series of complementation studies were performed with alleles of known mutants in the region. cDNAs from 5 genes in the region were obtained and *in-situ* hybridization of third instar larval eye imaginal discs was performed on *cn¹ bw¹* (wild-type) and *a69* mutants.

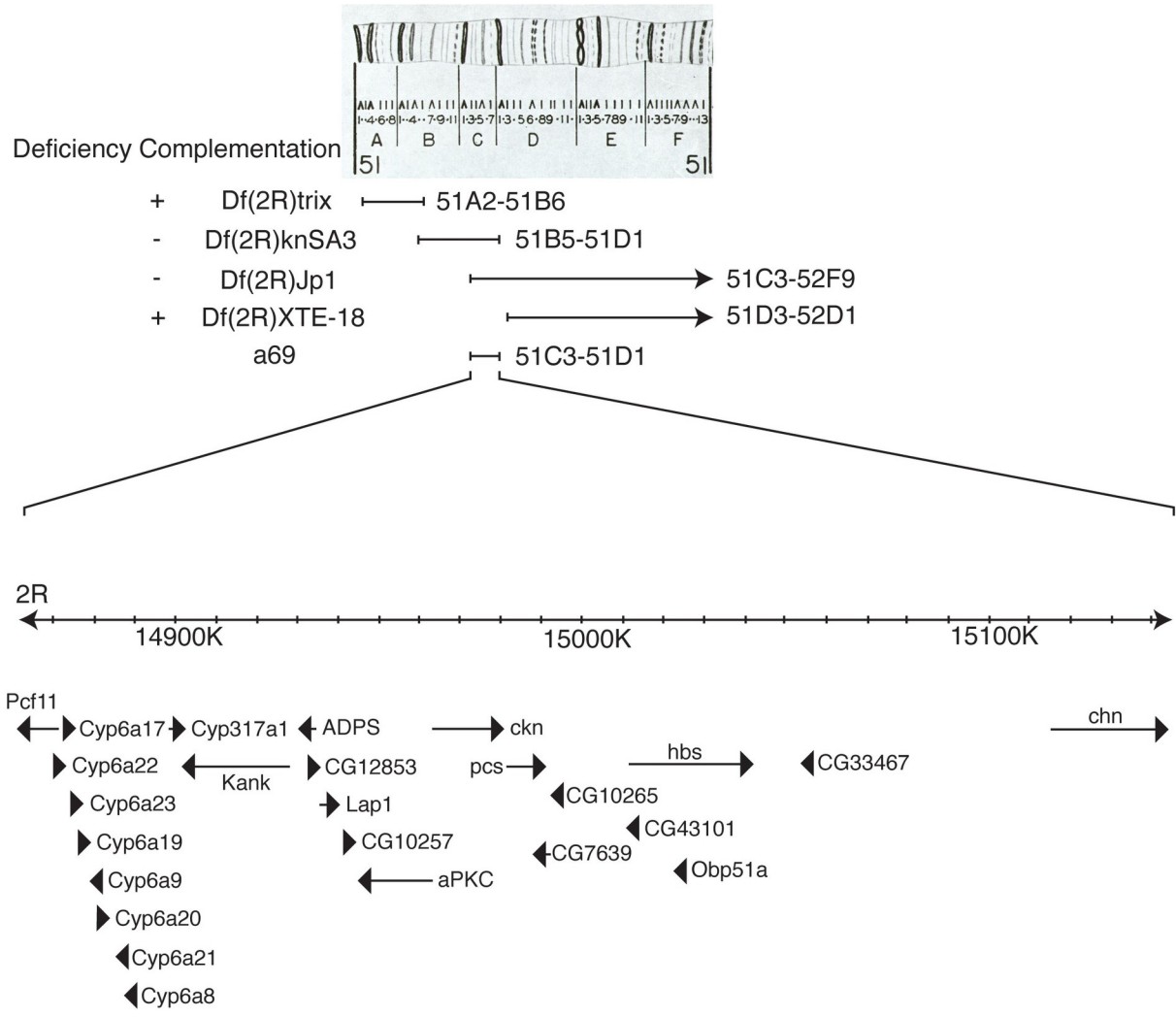

**Fig 4. Cytogenetic map, molecular map and deficiency complementation of *a69*.** The top panel shows the cytogenetic map of the 51 region of chromosome 2R [81], used with permission. Diagramed below are the deleted regions of deficiency strains tested, the corresponding molecular map and identified protein coding genes in the region. Arrows or arrowheads indicate the orientation of gene transcription and arrow or arrowhead length corresponds to gene length at the scale indicated (K, kilobase). Data obtained from Flybase version FB2018_01 [71].

In each case the expression pattern of the gene was not substantially disrupted in *a69* mutants, suggesting that the phenotype is not due to the disruption of patterned mRNA expression of these genes in the 3rd instar eye-antennal disc (**Fig 5**). *hibris* (*hbs*) was expressed strongly in the morphogenetic furrow and maintained weakly posteriorly, consistent with a previous report [32]. It was also expressed in the ocellar region and in the developing antenna. *parcas* (*pcs*, FBgn0033988) was expressed strongly in the morphogenetic furrow and in the antenna. *CG10265* (FBgn0033990) did not appear to be expressed in either the eye or antennal regions. *CG7639* (FBgn0033989) appeared to be weakly expressed in the region anterior to the morphogenetic furrow. *caskin* (*ckn*, FBgn0033987) was expressed anterior to the furrow and in the antenna.

We characterized *Rh5* and *Rh6* expression in animals heterozygous for *a69* and alleles of *Additional sex combs* (*Asx*, FBgn0261823), *atypical protein kinase C* (*aPKC*, FBgn0261854), *bocce* (*boc*, FBgn0011203), *charlatan* (*chn*, FBgn0015371), *Enhancer of GMR-sina 2–1* (*ES2-1*, FBgn0024358), *Hexokinase C* (*Hex-C*, FBgn0001187), *knot* (*kn*, FBgn0001319), *Regulatory particle non-ATPase 6* (*Rpn6*, FBgn0028689), *safranin* (*sf*, FBgn0003367), *Protein 1 of cleavage and polyadenylation factor 1* (*Pcf11*, FBgn0264962), *scab* (*scb*, FBgn0003326), and transposon insertions P{$A_{26}O_9$}1 (FBti0001751) and P{lacW}B6-2-25 (FBti0005748). All of these mutations complemented *a69*.

We obtained the following alleles of *hbs*: $hbs^{66}$ (FBal0239852), $hbs^{361}$ (FBal0130217), $hbs^{459}$, (FBal0130216), $hbs^{1130}$ (obtained from M. Baylies) and $hbs^{2593}$ (FBal0130218). With one exception, all of these alleles fail to complement *a69* (**Table 2**). Furthermore, $hbs^{361}$ homozygotes and heteroallelic combinations of all alleles show a substantial decrease in the proportion of *Rh5* expressing R8 photoreceptor cells. With a few exceptions, viable combinations of these alleles over deficiencies in the region show the same complementation pattern as the *a69* mutant (**S3 Table**).

Exon sequencing of the *hbs* gene failed to identify unique polymorphisms in the *a69* mutant that were absent in phenotypically wild type control strains. Nonetheless, given that the gene spans over 30 Kb including 24 Kb in the first intron, it seems likely that a mutation within a regulatory region of the *hbs* gene may be responsible for the hypomorphic *a69* phenotype. Based on the failure of complementation of *a69* by all but one allele of *hbs*, and the finding that all eleven heteroallelic combinations of four known alleles of *hbs* also display the *a69* phenotype (**Table 2**), we believe the data is consistent with *a69* being a *hbs* allele, $hbs^{a69}$. The two discrepancies to this conclusion, 1) complementation between *a69* and $hbs^{459}$, and 2) differences in complementation patterns of deficiencies (6 crosses out of 47, **S3 Table**) are consistent with intragenic (interallelic) complementation. This pattern of complex complementation, which has been described for numerous genes in *Drosophila*, is particularly common with alleles of intermediate phenotypic effects (e.g. hypomorphic alleles like *a69*), and may arise from differences in genetic background or complementation between alleles having defects in different functional regions of the gene [33–36]. One or more of these mechanisms are likely to underlie the complex complementation pattern observed in our experiments. Despite these discrepancies, the results with existing, molecularly characterized alleles of *hbs* clearly demonstrate that the *hbs* gene is required for the differentiation of R7 and R8 photoreceptor cells and the regulation of *Rh5* and *Rh6* opsin expression.

Because $hbs^{a69}$ is a hypomorph, subsequent genetic experiments were performed with the $hbs^{66}$ allele (FBal0239852), which has a stronger phenotype, is characterized at the molecular level and is available on the P{$ry^{+t7.2}$ = *neoFRT*}42D chromosome [37].

## *hibris* is expressed in the developing third instar eye imaginal disc

Consistent with *in situ* hybridization analyses (**Fig 5**) and previous studies [38], we find that the *hbs* protein is expressed in the developing third instar eye imaginal disc in the

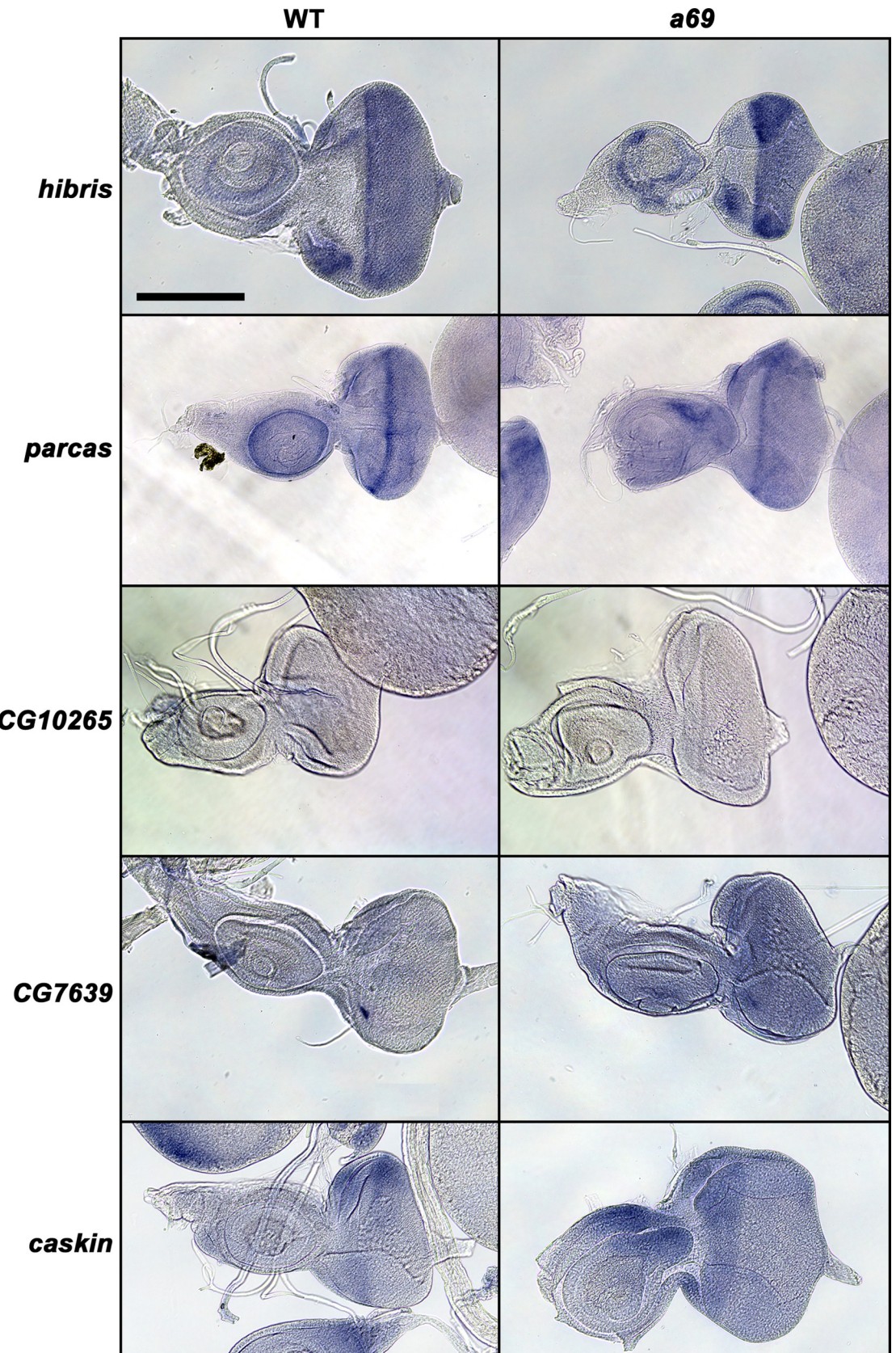

**Fig 5. *In situ* hybridization of *a69* candidate genes.** The panels show *in situ* hybridization of biotinylated reverse strand probes prepared from *hibris*, *parcas*, *CG10265*, *CG7639*, and *caskin* cDNA clones (rows) against wild type (WT) ($cn^1$ $bw^1$) (left column) or *a69* mutant (right column) eye-antennal imaginal discs. Scale bar = 100 μm for all panels.

morphogenetic furrow. The protein is found in preclusters of photoreceptor cells emerging from the morphogenetic furrow and ultimately in all photoreceptor cells (**Fig 6A and 6B**). *hbs* is expressed immediately anterior to, within, and posterior to the region of early *senseless* (*sens*, FBgn0002573) expression, which is expressed in R8 cells just posterior to the morphogenetic furrow (**Fig 6A–6C**).

### *hibris* is required in the retina for R7 and R8 cell differentiation

To assess the function of *hbs* in *Rh5* and *Rh6* expression in R7 and R8 photoreceptor cell patterning, we examined $hbs^{66}$ mosaic flies. We used the *ey-FLP* driver to generate homozygous mutant clones in the retina and optic lobes of animals that were heterozygous for $hbs^{66}$. We used a cell autonomous lethal mutation to generate large homozygous mutant clones and eliminate homozygous wildtype tissue, as described [39]. **Fig 7A** shows that loss of *hbs* in the retina and optic lobe leads to a dramatic decrease in *Rh5* expression and mispairing of *Rh3* and *Rh6* in adjacent R7 and R8 cells of individual ommatidia. This is in contrast to *Rh3*, *Rh5* and *Rh6* expression in a similarly FRT recombined clone of a wild type chromosome (**Fig 7B**, **S1 Fig**).

To further refine the spatial requirement for *hbs* in R7 and R8 photoreceptor cell differentiation and opsin gene expression we also generated mutant clones of $hbs^{66}$ with *ey3.5-FLP* [40]. *ey3.5-FLP* is a modified form of *ey-FLP* that efficiently induces clone formation in the third instar larval eye imaginal disc, but not in the lamina or medulla [40]. Retina specific clones generated with *ey3.5-FLP* also show a loss of *Rh5* expression along with increased mispairing of *Rh3* and *Rh6* (**Fig 7C**), as compared to an FRT recombined clone of a wild type chromosome (**Fig 7D**, **S1 Fig**). These results indicate that *hbs* is required in the retina for normal R7 and R8 photoreceptor cell differentiation and opsin gene expression.

**Table 2. Complementation crosses of *a69*, *hbs* alleles and *cn bw* control.**

| Genotype of Strains Crossed | $hbs^{66}$ | $hbs^{361}$ | $hbs^{459}$ | $hbs^{1130}$ | $hbs^{2593}$ | $cn^1$ $bw^1$ |
|---|---|---|---|---|---|---|
| *a69* | 6.6% (213) | 5.0% (337) | 22.9% (1164) | 10.4% (201) | 1.5% (455) | 25.7% (152) |
| | | | $p = 1.7 \times 10^{-4}$ | | | $p = 6.4 \times 10^{-4}$ |
| $hbs^{66}$ | | 2.4% (500) | 2.3% (399) | 2.1% (436) | 3.6% (419) | 31.8% (547) |
| | | | | | | $p = 3.9 \times 10^{-9}$ |
| $hbs^{361}$ | | 16.6% (404) | 3.3% (456) | 1.4% (358) | 2.7% (414) | 29.1% (320) |
| | | | | | | $p = 1.4 \times 10^{-6}$ |
| $hbs^{459}$ | | | | 3.9% (799) | 2.5% (651) | 33.3% (699) |
| | | | | | | $p = 1.3 \times 10^{-10}$ |
| $hbs^{1130}$ | | | | | 1.2% (326) | 26.8% (503) |
| | | | | | | $p = 5.2 \times 10^{-6}$ |
| $hbs^{2593}$ | | | | | | 30.7% (703) |
| | | | | | | $p = 8.5 \times 10^{-9}$ |

Statistical comparisons of strains were carried out as described in the Methods. Values shown are percentage of R8 cells expressing *Rh5* (number of ommatidia counted). The crossed alleles fail to complement *a69* and each other (shaded gray). Complementation in this table (unshaded) is an *Rh5*% significantly greater than *a69* homozygotes (12.7% (267)) by a one tailed test at the *p* value shown.

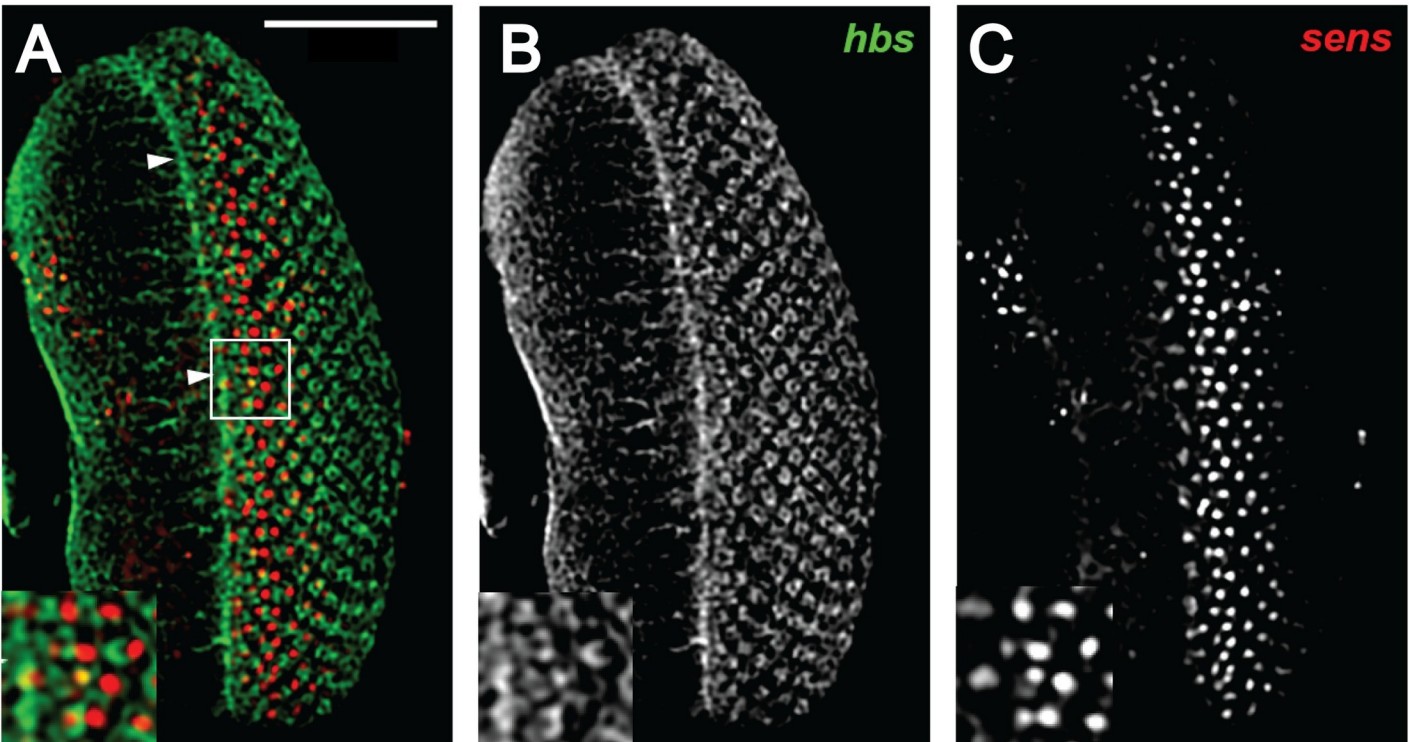

**Fig 6. *hibris* expression in the third instar larval eye imaginal disc.** Panel A shows a confocal microscopy flattened Z-stack series of *hibris* (*hbs*, green channel) and *senseless* (*sns*, red channel) double immunolabeling of a wild type (*cn¹ bw¹*) eye imaginal disc. Panel B shows the *hbs* labeling alone. Panel C shows the *sns* labeling alone. The morphogenetic furrow has moved from right (posterior) to left (anterior) and is located in the middle of the specimen (arrowheads, Panel A). The insets in each panel are a 2X magnification of the outlined region in Panel A. Scale bar = 50 μm for main panels, 25 μm for insets. The images are maximum intensity projections constructed from a series of z-stacks.

## Overexpression *of hibris* is sufficient to disrupt R7 and R8 cell differentiation

To determine whether ectopic expression of *hbs* is sufficient to induce the expression of *Rh5* in R8 photoreceptor cells, we over-expressed *hbs* using the GAL4-UAS system [41] and the *P {GAL4-ninaE.GMR}* driver (FBtp0001315), which drives transcription in the developing eye in all cell types posterior to the morphogenetic furrow [42, 43]. **Fig 8A** shows that overexpression of *hbs* leads to a large increase in *Rh5%* expression, demonstrating that *hbs* is sufficient to induce *Rh5* expression in many, but not all R8 photoreceptor cells. This occurs with a modest ~10% decrease in *Rh3-expressing* R7 cells (**Table 1**). To test whether this effect results from *hbs* acting on *Rh4* expressing R7 cells to inappropriately induce *Rh5* expression or from *hbs* acting directly on R8 cells, we overexpressed *hbs* in a *sevenless* (*sev*) mutant background that lacks R7 photoreceptor cells. **Fig 8B** shows that removal of R7 cells leads to a ~20% reduction in the number of *Rh5* expressing R8 cells, but still significantly more *Rh5* expression than is seen in *sev* mutants alone (**Table 1**). These results suggest that the ability of overexpressed *hbs* to induce *Rh5* expression in R8 cells is at least partially independent of the R7 photoreceptor cells and that *hbs* may act directly on, or in R8 cells.

## Materials and methods

### Stocks and genetics

Stocks were maintained in humidified incubators on cornmeal/molasses/agar media or standard cornmeal food with malt, and transferred on a rotating basis every three weeks as

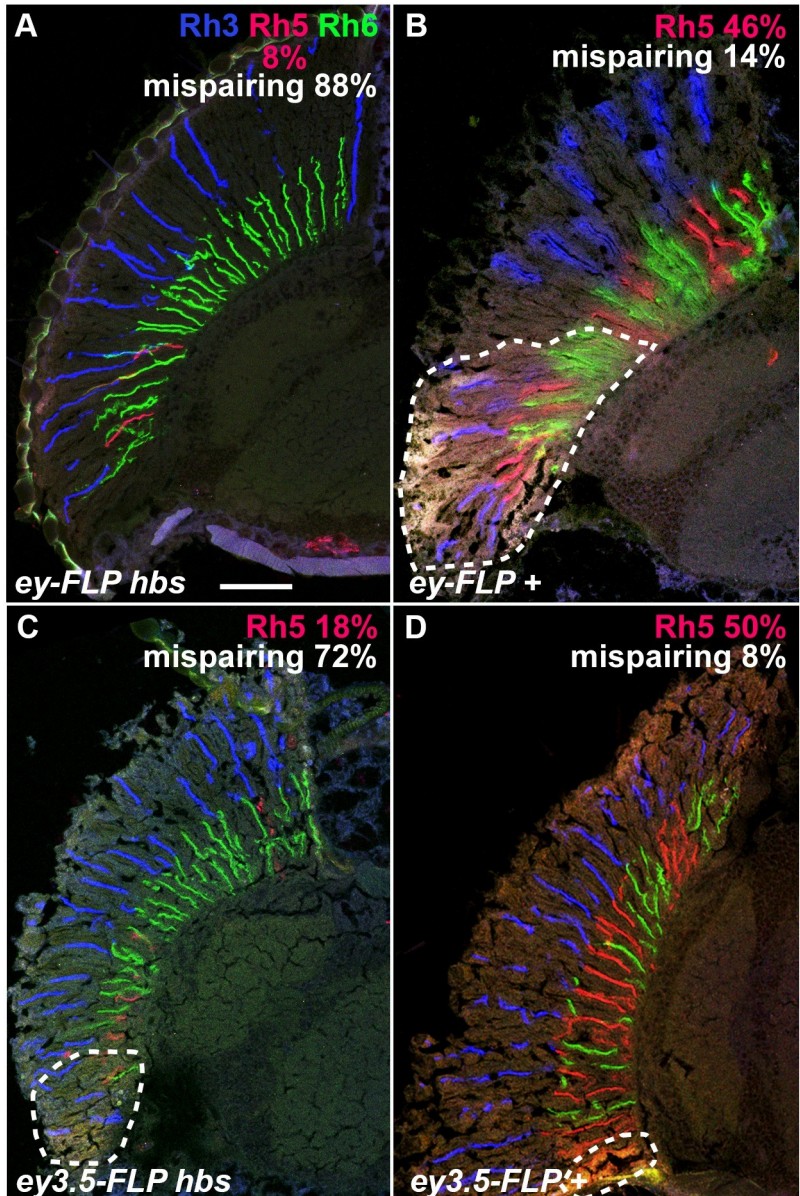

**Fig 7. Opsin expression in *hbs*[66] mutant and wildtype control flies.** Large FLP-FRT retinal clones were generated in the eye and optic lobes with *ey-FLP*, **A** and **B**, or in the retina alone with *ey3.5-FLP*. Homozygous *hbs*[66] mutant clones are shown in **A** and **C**. Homozygous wild-type control clones (+) are shown in **B** and **D**. Heterozygous tissue is marked with *w*[+] and outlined in panels **B, C** and **D**. *Rh3* (blue), *Rh5* (red) and *Rh6* (green) expression were detected by confocal microscopy with directly labeled monoclonal antibodies as described in **Materials and Methods**. *Rh5*% expression compared to *Rh6*, and *Rh3/Rh6* mispairing % compared to *Rh3/Rh5* are indicated in each panel. **A** *Rh5*% expression (n = 26), and *Rh3/Rh6* mispairing % (n = 16) are significantly different from **B** ($p$ = 0.017 and 0.0035, respectively (n = 13 and 7 for controls in **B**). **C** *Rh5*% expression (n = 28), and *Rh3/Rh6* mispairing % (n = 18) are significantly different from **D** ($p$ = 0.022 and 0.0022, respectively (n = 30 and 12 for controls in **D**). Scale bar = 50 μm for all panels. The quantitative data from this figure is shown in graph form in **S1 Fig**.

described [44–46]. *D. melanogaster* strains were obtained from individual laboratories or the Bloomington *Drosophila* Stock Center (BDSC). Genotypes were constructed using conventional genetic techniques, dominant markers and appropriate balancer chromosomes [45, 47].

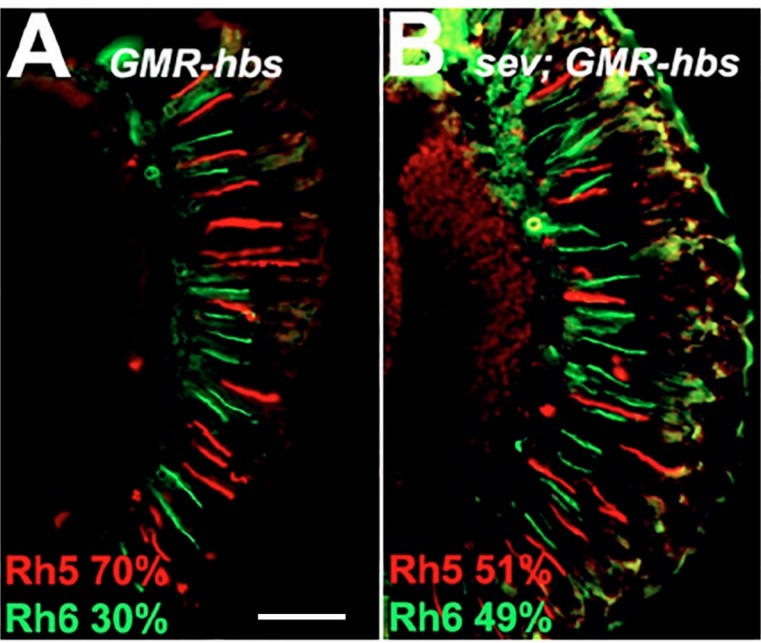

**Fig 8. Overexpression of *hibris* induces increased *Rh5* expression.** Over expression of *UAS-hbs* with the *GMR-GAL4* driver leads to an increase in *Rh5* (red) expression, panel A. Removal of R7 photoreceptor cells (*sevenless*[14] mutation) partially suppresses the effect, panel B. *Rh6* expression is shown in green. Scale bar = 50 μm for both panels. Statistics and comparisons with control strains are shown in Table 1.

A collection of new *P{etau-lacZ}* transposon (FBtp0001352) insertions was generated in our laboratory from *w; In(2LR)O, Duox^{Cy} P{etau-lacZ} / In(2LR)Gla, wg^{Gla-1}* (obtained from D. P. Smith) using *P{Δ2–3}99B* (FBtpi0000124) as a source of transposase [48]. Heterozygous strains carrying new transposon insertions (~1900) were screened for β-galactosidase expression in whole dissected adult heads [49]. Lines showing expression in the eye (retina and/or optic lobe) (323) were crossed to generate homozygous viable stocks. The collection was not retained long term and has since been discarded.

## Genotypes of animals shown in figures

**Fig 2A, 2B and 2C**: *w^{1118}*

**Fig 2D, 2E and 2F**: *w^{1118}*; *P{etau-lacZ}a69*

**Fig 5, Left column**: WT = *cn^1 bw^1*, **Right column**: *w^{1118}*; *P{etau-lacZ}a69*

**Fig 6**: *cn^1 bw^1*

**Fig 7A**: *w^{1118}/ y^{d2} w^{1118} P{ry^{+t7.2} = ey-FLP.N}2 P{GMR-lacZ.C(38.1)}TPN1; P{ry^{+t7.2} = neoFRT}42D hbs^{66}/ P{ry^{+t7.2} = neoFRT}42D P{w^{+t*} ry^{+t*} = white-un1}47A l(2)cl-R11^1*

**Fig 7B**: *w^{1118}/ y^{d2} w^{1118} P{ry^{+t7.2} = ey-FLP.N}2 P{GMR-lacZ.C(38.1)}TPN1; P{ry^{+t7.2} = neoFRT}42D / P{ry^{+t7.2} = neoFRT}42D P{w^{+t*} ry^{+t*} = white-un1}47A l(2)cl-R11^1*

**Fig 7C**: *w^{1118}/P{w^{+mC} = ey3.5-FLP.B}1, y^1 w^*; P{ry^{+t7.2} = neoFRT}42D hbs^{66}/ P{ry^{+t7.2} = neoFRT}42D P{w^{+t*} ry^{+t*} = white-un1}47A l(2)cl-R11^1*

**Fig 7D**: *w^{1118}/ P{w^{+mC} = ey3.5-FLP.B}1, y^1 w^*; P{ry^{+t7.2} = neoFRT}42D / P{ry^{+t7.2} = neoFRT}42D P{w^{+t*} ry^{+t*} = white-un1}47A l(2)cl-R11^1*

**Fig 8A**: *w^{1118}; P{GAL4-ninaE.GMR}12 / P{UAS-hbs.A}*

**Fig 8B**: *w^{1118} sev^{14}; P{GAL4-ninaE.GMR}12 / P{UAS-hbs.A}*

## Immunohistochemistry

10μm cryosections were prepared from heads of adult flies frozen in O. C. T. compound. The sections were fixed in 3% paraformaldehyde in Phosphate Buffered Saline (PBS pH 7.2) for 10 min, then permeabilized in cytoskeletal buffer (10mM Hepes [pH 7.4], 100 mM sucrose, 3 mM MgCl₂, 50 mM NaCl, 0.5% Triton X-100, 0.02% NaN₃) for 5 min. Specimens were incubated with the indicated primary antibody in antibody dilution buffer (3% Normal Goat Serum, 1 mg/ml BSA, and 0.03% Triton X-100 in PBS) for 1 hr at room temperature or overnight at 4°C. If necessary, secondary antibodies in antibody dilution buffer were incubated in an additional step. Between each step, slides were rinsed several times with PBS containing 0.01% saponin. Dissociated ommatidia were prepared from six animals. Eyes were cut from heads using 28 gauge needles in PBS. The retina, cornea +/- lamina tissue was shredded with needles, triturated 10 X with a 200 μL pipette tip and transferred to a microscope slide to dry at RT. Subsequent treatment was the same as cryosections. Primary antibodies were used at the following dilutions: directly conjugated mouse monoclonal anti-*Rh5* (Texas Red, 1:100, RRID: AB_2736994) and directly conjugated mouse monoclonal anti-*Rh6* (FITC, 1:100 RRID: AB_2736995) [50], rabbit polyclonal anti-*Rh4* (1:10, RRID:AB_2315271) [11, 50]. An additional reagent was prepared from purified (Cell Culture Company, LLC, Minneapolis, MN) mouse monoclonal anti-*Rh3* (RRID:AB_2315270). anti-*Rh3* was directly conjugated using Alexa Fluor™ 647 Protein Labeling Kit (Invitrogen, A20173) and used at 1:100 dilution. Secondary antibody used: goat anti-rabbit conjugated to rhodamine red (Jackson ImmunoResearch Laboratories, Inc. (West Grove, PA), 111-295-144). Immunofluorescence images were acquired with an Axioskop plus/AxioCamHRc (Carl Zeiss, Inc., Thornwood, NY) or by confocal microscopy using a Zeiss Pascal LSM (Carl Zeiss, Inc.) or Leica TCS SP5 (Leica Microsystems Inc., Buffalo Grove, IL).

Third instar eye-antennal discs were dissected in 1X PBS with 0.1% triton X-100 (PBT), fixed for 20 min at room temperature in 4% paraformaldehyde in 1X PBS, and washed three times for 5 min at room temperature with PBT. Discs were blocked with PBT supplemented with 1mg/mL BSA and 5% NGS for 1hr at room temperature before being incubated with primary antibodies diluted (as below) in PBT. After three 10 min washes, discs were incubated with secondary antibodies diluted 1:200 in PBT (see below) and once again washed three times for 10 min before being mounted in PermaFluor (ThermoFisher). Images were obtained on a Nikon A1R Confocal microscope and are constructed from a series of z-stacks as a maximum intensity projection. Primary antibodies used: guinea pig polyclonal anti-senseless (1:1000, [51]), rabbit polyclonal anti-hibris (1:400, AS-14, RRID:AB_2568633, [52]). Secondary antibodies used: goat anti-guinea pig conjugated to Alexa Fluor 568 and goat anti-rabbit conjugated to Alexa Fluor 488 (ThermoFisher A11075 and A11008, respectively).

## Statistical analysis

Comparisons of the proportions (percentages) of opsin expression in different genetic backgrounds were performed with a z-score and are shown in **Tables 1 and 2** legends for **Fig 7**, **S1–S3 Tables and S1 Fig** [53]. The *z*-score was calculated using the equation:

$$z = \frac{[\rho_2 - \rho_1] - \frac{1}{2}(1/n_1 + 1/n_2)}{\sqrt{\rho_{avg}q_{avg}(1/n_1 + 1/n_2)}}$$

*p₁* and *p₂* = proportions of marker expression in each of the two different genotypes under comparison. *n₁* and *n₂* = number of ommatidia counted for each genotype. *p_{avg}* = average

proportion for both genotypes combined. $q_{avg} = 1-p_{avg}$. The significance of the difference between the two proportions was determined from the normal distribution as a one- or two-tailed test. The 95% confidence interval of a proportion was calculated using the Wilson procedure without continuity correction [54, 55] using VasarStats [56].

### RNA *in situ* hybridization

Eye-antennal imaginal discs from third instar larvae were dissected in PBS, fixed in 50mM EGTA / 4% formaldehyde in PBS, rinsed in methanol, and stored in ethanol at -20˚. Discs were treated with ethanol/xylene (1:1), rinsed with ethanol, post-fixed in 5% formaldehyde in PBS plus 0.1% Tween (PBT), washed with PBT, and digested with Proteinase K (5 μg/ml). Tissue was post-fixed again and pre-hybridized in hybridization buffer (50% deionized formamide, 5XSSC, 1 mg/ml glycogen, 100 μg/ml salmon sperm DNA, 0.1% Tween) at 48˚C. Discs were hybridized overnight at 55˚C with 2 μl digoxigenin-labeled antisense RNA probe in 100 μl hybridization buffer. Probes were prepared from cDNA clones D1 [57], GH09755 (FBcl0125531), GM02985 (FBcl014202), LD18146 (FBcl0156485), LP09461 (Fbcl0187603) of genes *hbs*, *pcs*, *CG10265*, *CG7639* and *ckn*, respectively. The hybridized imaginal discs were washed extensively with hybridization buffer at 55˚C followed by PBT washes at room temperature. Discs were incubated with alkaline phosphatase-conjugated anti-digoxigenin antibody (1:2000, Roche Applied Science, Indianapolis, IN) overnight at 4˚C. Discs were washed with PBT and gene expression was visualized with staining solution (100mM NaCl, 50 mM $MgCl_2$, 100 mM Tris pH 9.5, 0.1% Tween) containing NBT/BCIP (Roche Applied Science). Stained imaginal discs were mounted and photographed using an Axioskop plus/AxioCamHRc (Carl Zeiss Inc.).

## Discussion

Here we describe the isolation and characterization of a novel allele of the *D. melanogaster* gene *hibris*, an evolutionarily conserved NPHS1 (nephrin) related IgSF member [58]. We show that *hibris* is required for the coordinated expression of opsin genes in adjacent R7 and R8 photoreceptor cells within the compound eye. Orthologues of this gene have been identified in many species, and numerous paralogues within species play diverse roles in organ system development and function [59]. Within the context of R7 and R8 photoreceptor cell differentiation and the regulation of opsin gene expression in the retinal mosaic, *hbs* is both required and partially sufficient for the expression of *Rh5* in R8 photoreceptor cells.

As noted briefly in the Introduction, the current model for the establishment of paired opsin gene expression in the R7 and R8 photoreceptors requires the type I activin receptor *baboon* (*babo*, FBgn0011300), bone morphogenetic protein type 1B receptor *thickveins* (*tkv*, FBgn0003716), transforming growth factor (TGF) beta type II receptor *punt* (*put*, FBgn0003169), many of their ligands, ligand processing convertases, and downstream effector enzymes [24]. In addition, the tumor suppressor kinase *warts* (*wts*, FBgn0011739), *hippo* kinase (*hpo*, FBgn0261456), *salvador* (*sav*, FBgn0053193), and *melted* (*melt*, FBgn0023001) a modulator of insulin/PI3K signaling [12], the *hpo* signaling cascade members *Merlin* (*Mer*, FBgn0086384), and *kibra* (*kibra*, FBgn0262127), the tumor suppressor *lethal (2) giant larvae* (*l(2)gl*, FBgn0002121) [22], and the transcription factors *ocelliless* (*oc*, FBgn0004102), *dorsal proventriculus* (*dve*, FBgn0020307) [60], *PvuII-PstI homology 13* (*Pph13*, FBgn0023489) [61] and *erect wing (ewg*, FBgn0005427) [62] are also required. Although not specifically tested in every case, all of these genes are thought to function cell autonomously within the R7 or R8 photoreceptor cells.

*hbs* is required in the retina for the induction of *Rh5* expression based upon our experiments making homozygous mutant clones with *ey3.5-FLP* (**Fig 7**). Subsequent studies will be required to place *hbs* within the large network of genes involved in R7 and R8 photoreceptor

cell differentiation and the inductive signal that is thought to coordinate opsin gene expression in adjacent R7 and R8 photoreceptor cells.

Traditionally, inductive processes are thought to occur between tissues or cells in which there is an inducer and a responder. Inductive signals are also often defined as instructive or permissive [63]. In the presence of an instructive interaction (i.e. from a R7p cell), the responder (R8) develops in a certain way (as a R8p cell expressing *Rh5*). By contrast, in the absence of the instructive interaction (R7y or R7 cells absent, e.g. *sev* mutants), the responder (R8) does not develop in a certain way, i.e. does not become R8p expressing *Rh5*, but rather becomes R8y and expresses *Rh6* instead as a default fate (with some exceptions [11]). If *hbs* played a formal instructive role in regulating the expression of *Rh5* in R8 photoreceptor cells, then we would expect that its expression throughout the retina (*GMR-Gal4; UAS-hbs*) would lead to expression of *Rh5* in all R8 photoreceptor cells even in the absence of R7 cells (**Fig 8B**). Although all *Rh5* is not expressed in all R8 cells in this experiment, it is far higher than in *sev* mutants alone [10, 11, 27, 28]. This demonstrates that *hbs* driven expression of *Rh5* in R8 photoreceptor cells is partially R7 cell independent and suggests that *hbs* may act on, or in R8 cells and play an instructive role in this process.

Alternatively, as a potentially permissive regulator of R8 photoreceptor cell differentiation, *hbs* may play a role in establishing the architecture of the developing eye. Perhaps loss of *hbs* disrupts cellular contacts that mediate signaling between R7 and R8. There is ample evidence for disruption of cone and pigment cell differentiation and eye roughening in *hbs* mutants [64, 65]. Furthermore, *hbs* and its binding partner *roughest* (*rst*) are known to have effects on axon guidance and synapse formation in the optic lobes [66–69]. Perhaps these interactions between retinal cells other than the R7 and R8 photoreceptor cells are responsible for some aspect of inductive signaling and expression of *Rh5* in R8p. Finally, perhaps the loss of *Rh5* expression in the *hbs* mutant eye reflects an inability to respond to the inductive signal, a loss of competence [70]. We previously suggested that *rhomboid* (*rho*, FBgn000463) and the *Epidermal growth factor receptor* (*Egfr*, FBgn0003731) may play a role in establishing competence of the R8 cell [27]. Thus, *hbs* could potentially play a permissive role in R7 and R8 differentiation.

Subsequent analysis of the role of *hbs* in R7 and R8 photoreceptor cell differentiation will require further identification of its specific interaction partners in this system in the retina, as well as the temporal requirement for its involvement in R7 and R8 cell differentiation. Ample resources are available including mutant strains [71], RNAi transgenics [72], and temporal and spatial mis-expression tools [73–77]. Despite these technical resources, defining the precise role of *hbs* in R7 and R8 differentiation will likely yield a complex system, reflecting coregulation of the IRM proteins [78], involvement of large complexes associated with scaffolding proteins [79], functional or genetic redundancy, compensation [80] and feedback.

## Supporting information

**S1 Fig. Quantification of opsin expression in *hbs*[66] mutant and wildtype control flies.** The bar graphs show quantification of the experiment in **Fig 7**. **A** *Rh5*% expression compared to *Rh6* is reduced in *ey-FLP hbs* and *eye3.5-FLP hbs* (first and third columns from the left), compared to *ey-FLP +* and *eye3.5-FLP +* controls (second and fourth columns from the left). **B** *Rh3/Rh6* mispairing % compared to *Rh3/Rh5* is increased in *ey-FLP hbs* and *eye3.5-FLP hbs* (first and third columns from the left), compared to *ey-FLP +* and *eye3.5-FLP +* controls (second and fourth columns from the left). Asterisks indicate *p*<0.05. Error bars indicate the 95% confidence intervals for the measured percentages. Additional quantitative data in **Fig 7 Legend**.
(TIF)

**S1 Table. Complementation of *a69* recombinant strains.** Recombinants described in Fig 3 were crossed to *a69* and the number of ommatidia counted expressing *Rh5* or *Rh6*, Total counted, and % *Rh5* are indicated in the table. Controls for comparison were homozygous *a69* mutants or *a69* / *w^{1118}* heterozygotes. Each recombinant strain was compared to both controls (right two columns) and was either not significantly different (NSD) or significantly different from (SDF) the indicated control at the *p* value stated. Statistical comparisons of strains were carried out as described in Materials and Methods. Controls are indicated at the bottom of the table. Recombinant strains having % *Rh5* values intermediate between wild type and mutant phenotypes, but statistically significantly different from both, are shaded.
(XLSX)

**S2 Table. Complementation of *a69* by deficiency strains.** A panel of thirty three deficiency stains were crossed to *a69* to test for complementation. The number of ommatidia counted expressing *Rh5* or *Rh6*, Total counted, and % *Rh5* are indicated in the table. The control for comparison was homozygous *a69* mutants. Compared to *a69* (right column) each deficiency over *a69* was either not significantly different (NSD) or significantly different from (SDF) *a69* at the *p* value stated. Statistical comparisons of strains were carried out as described in Materials and Methods. Values for the *a69* mutant are indicated at the bottom of the table. Deficiency strains failing to complement a69, which are not statistically significantly different from *a69*, or having a % *Rh5* significantly lower than *a69* are shaded.
(XLSX)

**S3 Table. Complementation of *hibris* alleles by deficiency strains.** A panel of seven deficiencies were crossed to *a69*, *hbs^{66}*, *hbs^{361}*, *hbs^{459}*, *hbs^{1130}*, *hbs^{2593}* and *cn^1 bw^1* to test for complementation of the *a69* mutant phenotype. The number of ommatidia counted expressing *Rh5* or *Rh6*, Total counted, and % *Rh5* are indicated in the table. The control for comparison was homozygous *a69* mutants. The deficiencies failed to complement the tested genotype (shaded rows) or complemented the tested genotype (white rows). Complementation was defined as significantly greater *Rh5*% than (SGT) *a69* homozygous mutant at the *p* value shown using a one-tailed test. Statistical comparisons of strains were carried out as described in Materials and Methods. Values for the *a69* mutant are indicated at the bottom of the table. Crosses having results that differed from *a69* are noted (Exceptions).
(XLSX)

**S4 Table. Strain information.** Includes recombination stocks, deficiencies and alleles that complement a69. Stock genetics, Flybase ID and RRID are listed where available.
(XLSX)

## Acknowledgments

We thank Mary Baylies, Ruben Artero, Gerry Rubin, Amy Tang, James Mohler, and Jeff Sekelsky for *D. melanogaster* stocks, Mary Baylies for the *hbs* D1 cDNA clone, Karl Fischbach for the rabbit anti-hbs antibody (AS-14), and Hugo Bellen for the guinea pig anti-sens antibody. Stocks obtained from the Bloomington Drosophila Stock Center (NIH P40OD018537) were also used in this study. We thank Natalia Toledo Melendez for technical assistance.

## Author Contributions

**Conceptualization:** Steven G. Britt.

**Formal analysis:** Hong Tan, Steven G. Britt.

**Funding acquisition:** Steven G. Britt.

**Investigation:** Hong Tan, Ruth E. Fulton, Wen-Hai Chou, Denise A. Birkholz, Meridee P. Mannino, David M. Yamaguchi, John C. Aldrich, Thomas L. Jacobsen.

**Supervision:** Steven G. Britt.

**Visualization:** Hong Tan, Ruth E. Fulton, Wen-Hai Chou, Denise A. Birkholz, Meridee P. Mannino, David M. Yamaguchi, John C. Aldrich, Thomas L. Jacobsen, Steven G. Britt.

**Writing – original draft:** Wen-Hai Chou, Steven G. Britt.

**Writing – review & editing:** Hong Tan, Ruth E. Fulton, Wen-Hai Chou, Denise A. Birkholz, Meridee P. Mannino, David M. Yamaguchi, John C. Aldrich, Thomas L. Jacobsen, Steven G. Britt.

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
