## [Decision Letter · Decision Letter 0]

4 May 2020

PONE-D-20-09777

*Drosophila* R8 photoreceptor cell subtype specification requires *hibris* and *Notch*.

PLOS ONE

Dear Dr. Britt

Thank you for submitting your manuscript to PLOS ONE. After careful consideration, we feel that it has merit but does not fully meet PLOS ONE’s publication criteria as it currently stands. Therefore, we invite you to submit a revised version of the manuscript that addresses the points raised during the review process.

Please find below the comments made by the reviewers on your paper. Although one of the reviewers agreed that the changes made to the original version addressed most of the problems, the second reviewer thinks that there are still some issues regarding interpretation of the data, especially the non-autonomous effect of hibris and the role of Notch. Please address all the reviewers’ comments, and pay particular attention to comments regarding the link with Notch. At this point I do not think you need to perform new experiments.

There are still typos in the manuscript for instance:

- Line 242 it says R7/RH8, it should be R7/R8

- Line 269 says icreased

We would appreciate receiving your revised manuscript within 4 weeks. To enhance the reproducibility of your results, we recommend that if applicable you deposit your laboratory protocols in protocols.io, where a protocol can be assigned its own identifier (DOI) such that it can be cited independently in the future. For instructions see: http://journals.plos.org/plosone/s/submission-guidelines#loc-laboratory-protocols

We look forward to receiving your revised manuscript.

Kind regards,

Carlos Oliva, PhD

Academic Editor

PLOS ONE

Reviewers' comments:

Reviewer's Responses to Questions

**Comments to the Author**

1. Is the manuscript technically sound, and do the data support the conclusions?

Reviewer #1: Yes

Reviewer #2: Yes

2. Has the statistical analysis been performed appropriately and rigorously? 

Reviewer #1: Yes

Reviewer #2: Yes

3. Have the authors made all data underlying the findings in their manuscript fully available?

Reviewer #1: Yes

Reviewer #2: Yes

4. Is the manuscript presented in an intelligible fashion and written in standard English?

Reviewer #1: Yes

Reviewer #2: Yes

5. Review Comments to the Author

Reviewer #1: In my opinion, the current version of the manuscript satisfies most of the issues raised by the reviewers during the previous round of revisions. However, some changes are still required prior to publication. Please find specific comments and suggestions below:

1) Materials and Methods section: At several occasions, the authors do not provide a detailed description of the methods used but rather refer to “previously described methods”. While this was considered as a normal practice until recently, this is not the case anymore. Readers should not have to quest a cascade of articles in order to figure out the exact procedures used in the paper. I strongly recommend that the authors add their own full description of the methods used in this study. Also, the paragraph on immunohistochemistry does not provide any indication about immunostainings on imaginal discs. The authors should complete it.

2) Figure 9: The authors substituted the control line (cn bw) used in the prior version of the manuscript by measures on ey flp; FRT GMR::GFP homozygous ommatidia. This new control presents a particularly high score of rh3 – rh6 mispairing (40%), much higher than the one of Hbs a69 homozygous mutants (table 1 – 25%) and of the ey-flp; FRT control flies used in figure 7 (14%). Such a strong effect is likely due to the homozygosity of the control line. A more appropriate control would have been to use ey flp; FRT 42A GMR::GFP / FRT 42A heterozygous flies. This would likely result in the reduction of this massive background effect. If the authors had the possibility to add this alternative control, this would be beneficial for the paper and make the results more solid. However, I do not think that this is absolutely mandatory, especially in the current confinement context. Alternatively, the authors should explicitly discuss the strong phenotype of their current control line in the manuscript.

3) Text, figures and tables details:

• line 220: A word – allele? mutation? - is missing in the sentence “We used a cell autonomous lethal *** to generate […]”.

• line 233: At the end of the paragraph describing the phenotype of hbs66 clones, the authors conclude by the following sentence “This phenotype is identical to hbsa69”. It is not clear what the authors mean by this sentence, especially at this place of the text, since they do not report any clonal analysis for the a69 allele. In addition, I would argue that, at this level of the paper, any reference to a69 is not so relevant anymore. I therefore recommend removing this sentence.

• Table 1: The authors should provide the percentage of Rh4/Rh5 mispairing for the gmr::hbs genotype.

• Figure 2 / 10: Please provide conventional scale bars, i.e. one scale bar per image corresponding to distances expressed as round numbers (10, 25, 50 µm), rather than 1 scale bar in one image and corresponding distances for the other images in the legend (with distances such as 11, 18 or 41 µm).

• Figure 6: legend: please replace “flattened confocal Z projections” by the exact denomination of the transformation you applied on the confocal Z stack (maximum intensity Z projection?).

• Figure 7: please also represent the data on a graph.

• line 262 - 264: Please clarify the following sentence: “there is a statistically significant increase in the percentage of mispaired Rh3-Rh6 expressing ommatidia ranging from 56 – 100% compared to the FRT control”. Indeed, the 56% - 100 % on figure 9 corresponds to the percentage of mispairing in mutant ommatidia and not the increase relative to the control.

• Figure 10 A: Some bars are missing in the histogram and should be added.

Reviewer #2: In this paper, the authors examine how color photoreceptor fates are coordinated in the fly eye. They examine homozygous viable enhancer trap lines and identify the a69 enhancer trap line which exhibits low Rh5 and miscoupling to Rh3-expressing R7s. They excised the P-element but still examined the phenotype. They mapped the mutation to Chromosome 2 and then used deficiency mapping to limit the mutation to a region with 25 known protein-coding genes. They test a number of genes for complementation and found that a69 often failed to complement alleles of hbs. They then tested hbs alleles for phenotypes and observed decreases in Rh5 expression. Overexpression of hbs in all photoreceptors increased Rh5 expression. Overexpression of hbs in all photoreceptors in a sevenless mutant lacking R7s also displayed an increase in Rh5. The authors then generated mutant clones and concluded that hibris was required in R7s and R8s. The authors concluded by using a ts allele of N to examine its role.

Overall, the identification of hbs as a new regulator of R8 subtype fate is exciting and well done. However, the conclusions of non-autonomy and a role for Notch are less clear.

I suggest that the authors address the following major issues:

1. The autonomy data is very complicated. The cell autonomous effect is very clear and strong. The authors observe expression in R8s and see a phenotype in R8 specific clones. Without evidence for expression in R7s, it is challenging to interpret the R7 specific clones. The authors should either 1. show that hibris is expressed in R7s, or 2. just show the R7+ R8+ vs. R7+ R8- data. Together, this clearly argues for a role in R8s. You can then move the other data to sup and say that it is suggestive of possible role in R7s.

2. The authors make the argument that GMR-hbs in a sev background suggests a non-autonomous role for hbs. However, this could simply be because GMR drives expression in R8s. I would remove this part of the argument. Generally, the authors should focus on the cell-autonomous role of hibris which is very strong.

3. The Notch data is extremely confusing and not convincing. I would highly suggest cutting it. In particular, the heat sensitivity of the system confounds the experiments.

4. Generally, the paper is very strong in its identification and characterization of the cell autonomous role of Hbs. I suggest that the authors change the focus of the paper to these very strong data and conclusions. These would include changes to the text including the title, intro, and discussion.

Minor comments

1. There are two copies of the manuscript including one with the track changes on. The authors should remove the second copy.

2. There are a number of references concerning R7 and R8 subtype fate specification that the authors missed. The authors should include these.

6. PLOS authors have the option to publish the peer review history of their article (what does this mean?). If published, this will include your full peer review and any attached files.

Reviewer #1: No

Reviewer #2: No

---

## [Author Response · Author response to Decision Letter 0]

24 Aug 2020

The reviewer’s comments are reproduced below with our point by point response.

Reviewer #1: In my opinion, the current version of the manuscript satisfies most of the issues raised by the reviewers during the previous round of revisions. However, some changes are still required prior to publication. Please find specific comments and suggestions below:

1) Materials and Methods section: At several occasions, the authors do not provide a detailed description of the methods used but rather refer to “previously described methods”. While this was considered as a normal practice until recently, this is not the case anymore. Readers should not have to quest a cascade of articles in order to figure out the exact procedures used in the paper. I strongly recommend that the authors add their own full description of the methods used in this study. Also, the paragraph on immunohistochemistry does not provide any indication about immunostainings on imaginal discs. The authors should complete it.

Author Response: Revised as requested. Additional detail is provided in the revised Materials and Methods. 

2) Figure 9: The authors substituted the control line (cn bw) used in the prior version of the manuscript by measures on ey flp; FRT GMR::GFP homozygous ommatidia. This new control presents a particularly high score of rh3 – rh6 mispairing (40%), much higher than the one of Hbs a69 homozygous mutants (table 1 – 25%) and of the ey-flp; FRT control flies used in figure 7 (14%). Such a strong effect is likely due to the homozygosity of the control line. A more appropriate control would have been to use ey flp; FRT 42A GMR::GFP / FRT 42A heterozygous flies. This would likely result in the reduction of this massive background effect. If the authors had the possibility to add this alternative control, this would be beneficial for the paper and make the results more solid. However, I do not think that this is absolutely mandatory, especially in the current confinement context. Alternatively, the authors should explicitly discuss the strong phenotype of their current control line in the manuscript.

Author Response: Revised as requested. The cell autonomy experiment is complicated by similar phenotypes in additional control experiments. Figure 9 has been deleted. 

3) Text, figures and tables details:

• line 220: A word – allele? mutation? - is missing in the sentence “We used a cell autonomous lethal *** to generate […]”.

Author Response: Revised as requested. The word “mutation” was added to the sentence.

• line 233: At the end of the paragraph describing the phenotype of hbs66 clones, the authors conclude by the following sentence “This phenotype is identical to hbsa69”. It is not clear what the authors mean by this sentence, especially at this place of the text, since they do not report any clonal analysis for the a69 allele. In addition, I would argue that, at this level of the paper, any reference to a69 is not so relevant anymore. I therefore recommend removing this sentence.

Author Response: Revised as requested. The sentence was deleted.

• Table 1: The authors should provide the percentage of Rh4/Rh5 mispairing for the gmr::hbs genotype.

Author Response: Revised as requested. This finding is not shown in Figure 8 or otherwise quantified and has been deleted from Table 1.

• Figure 2 / 10: Please provide conventional scale bars, i.e. one scale bar per image corresponding to distances expressed as round numbers (10, 25, 50 µm), rather than 1 scale bar in one image and corresponding distances for the other images in the legend (with distances such as 11, 18 or 41 µm).

Author Response: Revised as requested. Figure 2 has been revised. Figure 10 has been deleted.

• Figure 6: legend: please replace “flattened confocal Z projections” by the exact denomination of the transformation you applied on the confocal Z stack (maximum intensity Z projection?).

Author Response: Revised as requested.

• Figure 7: please also represent the data on a graph.

Author Response: Revised as requested. Bar graphs of this data are shown in S Fig 1.

• line 262 - 264: Please clarify the following sentence: “there is a statistically significant increase in the percentage of mispaired Rh3-Rh6 expressing ommatidia ranging from 56 – 100% compared to the FRT control”. Indeed, the 56% - 100 % on figure 9 corresponds to the percentage of mispairing in mutant ommatidia and not the increase relative to the control.

Author Response: Revised as requested. Figure 9 has been deleted along with the text noted above.

• Figure 10 A: Some bars are missing in the histogram and should be added.

Author Response: Figure 10 has been deleted as requested by Reviewer #2. 

Reviewer #2: In this paper, the authors examine how color photoreceptor fates are coordinated in the fly eye. They examine homozygous viable enhancer trap lines and identify the a69 enhancer trap line which exhibits low Rh5 and miscoupling to Rh3-expressing R7s. They excised the P-element but still examined the phenotype. They mapped the mutation to Chromosome 2 and then used deficiency mapping to limit the mutation to a region with 25 known protein-coding genes. They test a number of genes for complementation and found that a69 often failed to complement alleles of hbs. They then tested hbs alleles for phenotypes and observed decreases in Rh5 expression. Overexpression of hbs in all photoreceptors increased Rh5 expression. Overexpression of hbs in all photoreceptors in a sevenless mutant lacking R7s also displayed an increase in Rh5. The authors then generated mutant clones and concluded that hibris was required in R7s and R8s. The authors concluded by using a ts allele of N to examine its role.

Overall, the identification of hbs as a new regulator of R8 subtype fate is exciting and well done. However, the conclusions of non-autonomy and a role for Notch are less clear.

I suggest that the authors address the following major issues:

1. The autonomy data is very complicated. The cell autonomous effect is very clear and strong. The authors observe expression in R8s and see a phenotype in R8 specific clones. Without evidence for expression in R7s, it is challenging to interpret the R7 specific clones. The authors should either 1. show that hibris is expressed in R7s, or 2. just show the R7+ R8+ vs. R7+ R8- data. Together, this clearly argues for a role in R8s. You can then move the other data to sup and say that it is suggestive of possible role in R7s.

Author Response: Revised as requested. The cell autonomy experiment is complicated by similar phenotypes in additional control experiments, as noted above in response to Reviewer#1. Figure 9 has been deleted.

2. The authors make the argument that GMR-hbs in a sev background suggests a non-autonomous role for hbs. However, this could simply be because GMR drives expression in R8s. I would remove this part of the argument. Generally, the authors should focus on the cell-autonomous role of hibris which is very strong.

Author Response: Revised as requested. The description of the GMR-hbs experiment has been revised.

3. The Notch data is extremely confusing and not convincing. I would highly suggest cutting it. In particular, the heat sensitivity of the system confounds the experiments.

Author Response: Revised as requested. Figure 10 has been deleted.

4. Generally, the paper is very strong in its identification and characterization of the cell autonomous role of Hbs. I suggest that the authors change the focus of the paper to these very strong data and conclusions. These would include changes to the text including the title, intro, and discussion.

Author Response: Revised as requested. The title, introduction and discussion have been revised to reflect the altered focus of the manuscript.

Minor comments

1. There are two copies of the manuscript including one with the track changes on. The authors should remove the second copy.

Author Response: The instructions to authors request one version of revised manuscripts with the changes highlighted. This is noted in the title of the file.

2. There are a number of references concerning R7 and R8 subtype fate specification that the authors missed. The authors should include these.

Author Response: Revised as requested. Additional references have been cited.

---

## [Decision Letter · Decision Letter 1]

28 Sep 2020

*Drosophila* R8 photoreceptor cell subtype specification requires *hibris*.

PONE-D-20-09777R1

Dear Dr. Britt,

I am pleased to inform you that your manuscript has been judged scientifically suitable for publication and will be formally accepted for publication once it meets all outstanding technical requirements.

Kind regards,

Carlos Oliva, PhD

Academic Editor

PLOS ONE

Additional Editor Comments (optional):

Reviewers' comments:

Reviewer's Responses to Questions

**Comments to the Author**

1. If the authors have adequately addressed your comments raised in a previous round of review and you feel that this manuscript is now acceptable for publication, you may indicate that here to bypass the “Comments to the Author” section, enter your conflict of interest statement in the “Confidential to Editor” section, and submit your "Accept" recommendation.

Reviewer #1: All comments have been addressed

Reviewer #2: All comments have been addressed

2. Is the manuscript technically sound, and do the data support the conclusions?

Reviewer #1: Yes

Reviewer #2: Yes

3. Has the statistical analysis been performed appropriately and rigorously? 

Reviewer #1: Yes

Reviewer #2: Yes

4. Have the authors made all data underlying the findings in their manuscript fully available?

Reviewer #1: Yes

Reviewer #2: Yes

5. Is the manuscript presented in an intelligible fashion and written in standard English?

Reviewer #1: Yes

Reviewer #2: Yes

6. Review Comments to the Author

Reviewer #1: (No Response)

Reviewer #2: The authors have satisfactorily addressed the reviewers' concerns and the manuscript is ready for publication.

7. PLOS authors have the option to publish the peer review history of their article (what does this mean?). If published, this will include your full peer review and any attached files.

Reviewer #1: No

Reviewer #2: No

---

## [Editor Report · Acceptance letter]

5 Oct 2020

PONE-D-20-09777R1 

*Drosophila* R8 photoreceptor cell subtype specification requires *hibris*. 

Dear Dr. Britt:

I'm pleased to inform you that your manuscript has been deemed suitable for publication in PLOS ONE. Congratulations! Your manuscript is now with our production department. 

Kind regards, 

on behalf of

Dr. Carlos Oliva 

Academic Editor

PLOS ONE